# Comprehensive Analysis of Phenolic Constituents, Biological Activities, and Derived Aroma Differences of *Penthorum chinense* Pursh Leaves after Processing into Green and Black Tea

**DOI:** 10.3390/foods13030399

**Published:** 2024-01-26

**Authors:** Zhuoya Xiang, Boyu Zhu, Xing Yang, Junlin Deng, Yongqing Zhu, Lu Gan, Manyou Yu, Jian Chen, Chen Xia, Song Chen

**Affiliations:** 1Institute of Agro-Products Processing Science and Technology (Institute of Food Nutrition and Health), Sichuan Academy of Agricultural Sciences, 60 Shizishan Road, Chengdu 610066, China; xiangzhuoya2015@163.com (Z.X.); 18227551186@139.com (B.Z.); yangyx0816@163.com (X.Y.); 13547499730@163.com (J.D.); yongq293@163.com (Y.Z.); gravity_lost@163.com (L.G.); jiangymy518@163.com (M.Y.);; 2Gucui Biotechnology Co., Ltd., Luzhou 646500, China; chensong@163.com

**Keywords:** *P. chinense* leaves, processing, phenolic constituents, volatile compounds, biological activities

## Abstract

*Penthorum chinense* Pursh (Penthoraceae) is a traditional herb used in Miao medical systems that is also processed into foods (e.g., tea products) in China. Different processing methods significantly affect the volatile compounds, phenolic constituents, and biological activities. This study aimed to produce *P. chinense* green tea leaves (GTL), black tea leaves (BTL), and untreated leaves (UL) to investigate differences in their flavor substances, functional components, antioxidant activity, alcohol dehydrogenase (ADH) activity, and acetaldehyde dehydrogenase (ALDH) activity. The results showed that 63, 56, and 56 volatile compounds were detected in UL, GTL, and BTL, respectively, of which 43 volatile compounds were identified as differential metabolites among them. The total phenolic content (97.13–179.34 mg GAE/g DW), flavonoid content (40.07–71.93 mg RE/g DW), and proanthocyanidin content (54.13–65.91 mg CE/g DW) exhibited similar trends, decreasing in the order of UL > BTL > GTL. Fourteen phenolic compounds were determined, of which gallic acid, (−)-epicatechin, and pinocembrin 7-*O*-glucoside showed a sharp decrease in content from UL to BTL, while the content of pinocembrin 7-*O*-(3″-*O*-galloy-4″, 6″-hexahydroxydiphenoyl)-glucoside and pinocembrin significantly increased. GTL showed better DPPH/ABTS^·+^ scavenging ability and ferric-reducing ability than UL. The ADH and ALDH activities decreased in the order of GTL > UL > BTL. Therefore, tea products made with *P. chinense* leaves contained an abundance of functional compounds and showed satisfactory antioxidant and hepatoprotective activities, which are recommended for daily consumption.

## 1. Introduction

*Penthorum chinense* Pursh (family Penthoraceae), referred to as “Gan Huang Cao” in Chinese, is a traditional herb in the Penthoraceae family. It is widely distributed in China and used in the Miao medical systems, especially against liver diseases including cholecystitis, alcoholic liver diseases, and infectious hepatitis [1,2]. Furthermore, traditional Chinese medical preparations under the trademark of “Gansu” for the treatment of liver diseases contain only the extract of *P. chinense.* China is one of the primary producers of *P. chinense*, in which Sichuan Province has the largest area dedicated to *P. chinense* cultivation. The entire plant is often used in medicine, with the stems commonly used for preparations [3]. However, the utilization of *P. chinense* in medicine is limiting, although it is an edible plant used as a vegetable by the locals. Thus, it is necessary to investigate the comprehensive development and utilization of *P. chinense* to provide practical and economic guidance for its utilization.

Tea is a popular and healthy beverage in modern life. With the increasing varieties of food, current research has considered novel tea products made from new materials and resources. Modern research confirmed the negligible toxicity of *P. chinense* as an edible plant [4], which is also rich in bioactive polyphenols, including flavonoids, flavonoid glycosides, phenylpropanoids, phenols, and steroids [5]. Studies have shown that gallic acid and quercetin exhibit anti-HBV (viral hepatitis type B) activity and liver protection ability against alcohol liver and fatty liver [6,7]. Processing the leaves of the crops into tea is a common practice that can enhance their economic benefits. Similar studies have been performed on mulberry (*Morus alba* L.) leaf, Dang Shen (*Codonopsis pilosula*) leaf, and mango (*Mangifera indica* L.) leaf tea [8,9,10]. Besides, processing may promote the transformation of various internal bioactive compounds in herb, producing various active metabolites [11]. Moreover, study indicated that the biotransformation of fermented process has been shown to effectively improve the chemical composition of raw materials and enhance biological activity [8]. However, little information exists regarding the transformation of the chemical composition and biological activities of *P. chinense* leaves during the processing into two tea products through unfermented and fermented processes.

The objectives of this study were to compare different processing method between *P. chinense* leaves green tea and *P. chinense* leaves black tea, and their phenolic composition, volatile flavor compounds, antioxidant activity, and hepatoprotective activity of products were analyzed. These findings could provide guidance on the utilization of *P. chinense* leaves under practical production conditions and may improve our understanding of phenolic compound accumulation patterns in *P. chinense* leaves tea related to their processing characteristics.

## 2. Materials and Methods

### 2.1. Chemicals and Reagents

Na_2_CO_3_, Al(NO_3_)_3_, formic acid, NaNO_2_, HCl, ethanol, Folin–Ciocalteu phenol reagent, and Trolox were obtained from Chengdu Kelong Chemical Reagent Works (Chengdu, China). 1,1-Diphenyl-2-picrylhydrazyl (DPPH^·^, >99.7%), 2,2′-azino-bis(3-ethylbenzothiazoline-6-sulfonic acid) diammonium salt (ABTS^·+^, >99.7%), and chromatographic-grade methanol and acetonitrile were purchased from American Sigma (St. Louis, MO, USA). Standard compounds including (+)-catechin, (−)-epicatechin, rutin, afzelin, astragaline, kaempferol-3-*O*-rutinoside, kaempferol, quercetin, isoquercitrin, pinocembrin, pinocembrin 7-O-glucoside, thonningianin A, gallic acid, and pinocembrin 7-*O*-(3″-*O*-galloy-4″,6″-hexahydroxydiphenoyl)-glucoside were purchased from Shanghai Yuanye Bio-Technology Co., Ltd. (Shanghai, China). Alcohol dehydrogenase (265 units/mg) from saccharomyces cerevisiae was purchased from Sigma-Aldrich, Co. (St. Louis, MO, USA).

### 2.2. Preparation of Tea Samples

Untreated *P. chinense* leaves (UL) were collected from Huangjing town of Gulin County, China (location 28° N and 105° E) in August 2022. The leaves were freeze-dried immediately, ground to 60 mesh, sealed, and stored at −20 °C for testing.

*P. chinense* green tea leaves (GTL): Briefly, fresh leaves were spread out indoors for 2 h and we then performed stir-fixation at 320 °C for 2–3 min using a fixation machine (Sunyoung Machinery Co., Ltd., Quzhou, China) for deactivation of the enzymes. A carding machine (Sunyoung Machinery Co., Ltd., Quzhou, China) was used to shape the tea shoots at 250–280 °C for 8–10 min, which were subsequently dried at 100 °C for 5 min and then at 68 °C for 30 min.

*P. chinense* black tea leaves (BTL): Briefly, fresh leaves were spread out indoors for 2 h, and then pan-fired at 320 °C for 2–3 min using a fixation machine. A carding machine (Sunyoung Machinery Co., Ltd., Quzhou, China) was used to shape the tea shoots for 30 min. They were then fermented in a pile for 6 h at 30 °C. After drying at 100 °C for 5 min and at 68 °C for 30 min, the tea leaves were allowed to cool to room temperature. All tea samples were preserved in aluminum foil bags and stored at −20 °C for further analysis.

### 2.3. Analysis of Volatile Components

The volatile components in the *P. chinense* samples were determined by headspace solid-phase microextraction (HS-SPME). Briefly, 1.3 g of sample powder was added to a 20 mL headspace vial, which was then heated at 50 °C for 15 min for equilibration. Then, SPME fibers were coated with 50/30 μm DVB/CAR (Supelco Corporation, Bellefonte, PA, USA) and inserted into headspace vials for extraction at 50 °C for 30 min. Subsequently, the fibers were desorbed at 260 °C for 5 min.

The volatiles were separated by gas chromatography–mass spectroscopy (GC-MS) using a gas chromatography system (7890 B, Agilent Corporation, Santa Clara, CA, USA) equipped with a DB-wax column (30 m × 0.25 mm × 0.25 μm, Agilent Corporation) with an initial column temperature of 40 °C for 5 min that was then increased to 250 °C at 5 °C/min and held for 3 min. The helium flow rate was 1.0 mL/min, the electron ionization energy was 70 eV, and the mass scanning range was 20–400 amu. The raw data were identified by comparing with the NIST 17 database using MSD ChemStation software (F.01.03.2357, Version 2.2).

### 2.4. Sample Extraction

Briefly, 1 g of dry sample powder was homogenized with 8 mL of 80% ethanol. The mixture was then sonicated for 30 min at 40 °C, and the extract was separated by centrifugation (8000 rpm, 10 min). The supernatant was collected, and the extraction procedure was repeated twice. The extracts were fixed to 25 mL and then filtered with a 0.22 μm syringe filter. Each sample was prepared in triplicate.

### 2.5. Total Phenolic Content (TPC)

The TPC was determined using the Folin–Ciocalteu colorimetric method. First, 20 μL of Folin–Ciocalteu reagent was added to 20 μL of extracts and left to stand for 5 min. Then, 160 μL of 5% Na_2_CO_3_ was added and reacted for 60 min at room temperature. The absorbance of the mixture was determined at 765 nm, and the results were expressed as mg (gallic acid equivalent, GAE)/g (dry weight, DW).

### 2.6. Total Flavonoid Content (TFC)

The TFC was determined by an aluminum chloride colorimetric assay. First, 15 μL of 5% NaNO_2_ was mixed with 20 μL of extracts and reacted for 6 min. Then, 10 μL of 10% Al(NO_3_)_3_ was added and reacted for 5 min. Finally, 30 μL of 1 M NaOH was added, and the sample was determined at 510 nm. The results were expressed as mg (rutin equivalent, RE)/g (DW).

### 2.7. Total Proanthocyanidin Content (TPAC)

The TPAC was determined according to Xiang et al. [12]. Briefly, 20 μL of extract was mixed with 100 μL of 1% vanillin, and 100 μL of 4% HCl was added. The mixture was kept at 37 °C for 20 min before measurement at 500 nm. The results were expressed as mg (catechin equivalent, CE)/g (DW).

### 2.8. Phenolic Composition

The extracts were analyzed using an Agilent LC-1290 HPLC system (Agilent Corporation, Santa Clara, CA, USA). Chromatographic separation was conducted on an InfinityLab Poroshell 120 PFP column (4.6 × 100 mm, 2.7 µm particle size, Agilent Corporation). The mobile phase consisted of solvent A (0.1% formic acid in water) and solvent B (acetonitrile). The gradient program was set as follows: 0–10 min (5–10% B), 10–15 min (10–20% B), 15–20 min (20–35% B), 20–30 min (35–75% B), 30–32 min (75–95% B); flow rate, 0.8 mL/min; column temperature, 30 °C; detection wavelengths, 280 and 350 nm; injection volume, 2 µL. The results were expressed as µg (phenolics)/g (DW).

### 2.9. Antioxidant Activity

The antioxidant activity was evaluated using two radicals (DPPH^·^ and ABTS^·+^) and a ferric-reducing ability (FRAP) assay. For the DPPH^·^ free radical scavenging ability, 100 µL of extract was combined with 100 µL of DPPH^·^ solution and then incubated in the dark for 30 min before the absorbance was measured at 517 nm. For the ABTS^·+^ radical scavenging ability, 160 μL of ABTS^·+^ solution, and 40 μL of extract were reacted in the dark for 6 min, after which the absorbance at 734 nm was measured. For the ferric-reducing ability, 30 μL of extract was mixed with 256 μL of FRAP reagent. This mixture was kept at room temperature for 30 min before the absorbance was measured at 593 nm. Vitamin E (VE) served as the positive control. The inhibitory concentration at 50% (IC_50_) reflected the antioxidant concentration of the tested samples needed to neutralize 50% of the initial concentration of free radicals.

### 2.10. ADH and ALDH Activities

The ADH activity was measured by the Valle and Hoch method [13]. In brief, 1.5 mL of pyrophosphate buffer (32 mM, pH 8.8), 1 mL of NAD^+^ (27 mM), 0.5 mL of ethanol (11.5%), and 0.1 mL of sample (25 μg/mL) were mixed at 25 °C for 5 min. Then, 0.1 mL of ADH (0.25 U/mL) was added, and the absorbance was immediately read at 340 nm for 5 min until the increase in absorbance per minute reached a stable value.

The ALDH activity was determined using a modified Blair and Bodley method [14]. In brief, 1.5 mL of phosphate buffer (pH 8.0, 124 mM), 0.5 mL of acetaldehyde (6.4 mM), 1 mL of NAD^+^ (16 mM), and 0.1 mL of sample (25 µg/mL) were mixed at 25 °C. Then, 0.1 mL of ALDH (10 mM) was added to initiate the reaction. The absorbance was immediately measured at 340 nm and measured again after the mixture was warmed at 37 °C for 5 min until the increase in absorbance per minute reached a stable value.
*Q* = (*E*_1_ − *E*_0_)/*E*_0_ × 100%
where *Q* is the activation rate of ADH or ALDH (%), *E*_1_ is the enzyme activity in the sample solution (U/mg), and *E*_0_ is the enzyme activity in the blank solution (U/mg).

### 2.11. Statistical Analysis

All experiments were determined in triplicate replicates unless specified otherwise. The data were calculated using analysis of variance (ANOVA) followed by Duncan’s test (*p* < 0.05). All statistical analyses were calculated using SPSS 22.0 software (version 22.0, SPSS Inc., Chicago, IL, USA). Heat maps and partial least-squares discriminant analysis (PLS-DA) was performed using MetaboAnalyst V 5.0 (www.Metaboanalyst.ca, accesed on 23 November 2023).

## 3. Results and Discussion

### 3.1. Volatile Flavor Compounds

Aroma is an important factor used to evaluate the quality of tea products, with different processing methods having significant effects on tea aroma. The volatile flavor of *P. chinense* leaves and their tea products were analyzed, and the results are shown in Figure 1A, Appendix A. In UL, 63 unique volatile flavor compounds were identified, comprising 12 alcohols, 11 aldehydes, 13 esters, 5 acids, 8 ketones, 8 hydrocarbons, 3 oxacycles, and 3 other substances. In GTL, 56 volatile flavor compounds were identified, comprising 12 alcohols, 16 aldehydes, 9 esters, 6 acids, 5 ketones, 3 hydrocarbons, 3 oxacycles, and 2 other substances. In BTL, 56 volatile flavor compounds were identified, comprising 12 alcohols, 16 aldehydes, 10 esters, 3 acids, 7 ketones, 3 hydrocarbons, 4 oxacycles, and 1 other substance. Alcohols, aldehydes, and esters were the main volatile compounds found in *P. chinense* and its products, similar to Kombucha tea [15].

The total number of flavor substances in UL was more than that of BTL or GTL. Butyrolactone; ammonium acetate; bicyclo[2.2.1]heptan-2-one; terpinen-4-ol; benzoic acid, methyl ester; thujone; hexadecanoic acid, ethyl ester; benzeneacetic acid, ethyl ester; and tetradecanoic acid, ethyl ester were only found in UL (Figure 2) and could be the representative flavor substances of *P. chinense* leaves. Among them, thujone is a volatile monoterpene ketone of traditional medicinal plants and a γ-aminobutyric acid regulator [16]. For tea products, high-temperature roasting or fermenting is an important processing stage used to create new flavor substances. BTL formed by fermented tea processing contained slightly lower amounts of volatile flavor substances than GTL formed through non-fermentation tea processing. BTL had a higher aroma concentration than GTL due to its complex processing method [17]. These substances were further fermented during fermentation, resulting in the production of more flavor substances, such as 2-pentanone, formic acid, butyl ester, 1-hexanol, 2-hexen-1-ol, 1-penten-3-one, and *cis*-3-hexenyl *cis*-3-hexenoate. Therefore, fermentation could improve the quality and quantity of volatile flavor components, as related to the study by Pripdeevech and Machan [18].

PCA score analysis showed that the nine samples divided into UL, GTL, and BTL were clustered well (Figure 1B), with PC1 and PC2 accounting for 50.9% and 22.9% of the total variance, respectively, and with PC1 and PC2 combined accounting for 73.8% of the total variance. The improved classification was likely due to differences in the volatile flavor level related to the processing method [17]. PLS-DA was used to investigate the distribution differences of volatile components between the different samples. The PLS-DA score plot (R2: 0.893, Q2: 0.758) demonstrated that UL, GTL, and BTL were well discriminated by the volatile metabolites identified. Among the compounds identified, 43 volatile compounds were selected as key differential markers in the PLS-DA model with a threshold of 1.0 and *p*-values less than 0.05 [19], which contributed significantly to the aroma of the constituents in *P. chinense* leaves and their tea products. Zhang et al. [20] analyzed the flavor substances of black, green, and white tea. From a total of 46 detected flavor substances, it was suggested that phenylethanol, decanol, octanoic acid, ethyl octanoate, and ethyl decanoate could be used as the representative flavor substances of Kombucha tea. Clustering heat map analysis was subsequently used to provide detailed information on the dynamic changes in the differential volatile compounds of different samples. The pattern of changes in concentration showed a regular transformation in different samples (Figure 2). More than half of the esters, ketones, and acids present in UL contributed to its fresh, sweet, sour, and floral odors due to the high content of cyclohexene, (*S*)-1-methyl-4-(1-methylethenyl)-, and ammonium acetate, which differed from its tea products. Upon processing, the content of some of the original aromatic compounds in the fresh leaves decreased, while fermentation-induced aromatic compounds like (*Z*)-4-heptenal, *cis*-3-hexenyl *cis*-3-hexenoate, and 1-penten-3-one, gradually increased in content, which contributed to their grassy and fatty odor. Above all, 16 differential volatile components had a higher intensity in GTL and BTL than UL, indicating that tea processing can promote the transformation of various active metabolites.

### 3.2. Differences in TPC, TFC, and TPAC

As shown in Table 1, the TPC, TFC, and TAPC showed a similar trend, decreasing in the order of UL > BTL > GTL. Polyphenols, specifically flavonoids and flavonoid glycosides, are the main active components in *P. chinense* Pursh leaves. UL had the highest polyphenol (179.34 mg/g DW), flavonoid (71.93 mg/g DW), and proanthocyanidin content (65.91 mg/g DW) among the three types of *P. chinense* Pursh leaves. From fresh leaves to green and black tea, the total polyphenol contents decreased by 22.55% and 45.84%, the total proanthocyanidin contents decreased by 16.75% and 17.87%, and the total flavonoid contents decreased by 12.39% and 44.29%, respectively. During the processing of tea, high-temperature roasting or fermenting accelerated the transformation of polyphenols and flavonoids, resulting in a sharp decrease in content, consistent with previous research [8]. In addition, the TPC and TFC of GTL were significantly higher than those of BTL. Green tea is processed at high temperatures to inhibit enzymatic oxidation and prevent the oxidation of polyphenols. However, in the fermentation of BTL, phenolic substances are continuously consumed due to the oxidation, polymerization, and condensation of polyphenols. Therefore, BTL had a lower polyphenol content than GTL. Studies also indicated that the bitterness and astringency of tea decreased with the polyphenol content after high-temperature treatment [21], making BTL more suitable as a tea drink.

### 3.3. Differences in Individual Phenolic Compounds

Earlier reports revealed that *P. chinense* contains many active polyphenols including flavonoids, phenylpropanoids, organic acids, and phenols [3]. We further determined the contents of 14 polyphenols, as shown in Table 2, comprising two flavanols ((+)-catechin and (−)-epicatechin), seven flavonols (rutin, afzelin, astragaline, kaempferol, quercitrin, kaempferol-3-*O*-rutinoside and isoquercitrin), three flavanones (pinocembrin, pinocembrin 7-*O*-(3″-*O*-galloy-4″,6″-hexahydroxydiphenoyl)-glucoside and pinocembrin 7-O-glucoside), one dihydrochalcone (thonningianin A), and one phenolic acid (gallic acid).

The contents of individual phenolic compounds varied in each sample. The primary phenolic compound contents of UL, GTL, and BTL were similar, while components such as gallic acid, (−)-epicatechin, and pinocembrin 7-O-glucoside exhibited significant differences and showed a sharp decreasing trend. In contrast, the content of pinocembrin 7-*O*-(3″-*O*-galloy-4″,6″hexahydroxydiphenoyl)-glucoside and pinocembrin significantly increased. In the production process of tea, heat and fermentation accelerated the transformation of polyphenols and flavonoids, resulting in a sharp decrease or increase in content. This change was consistent with previous tea processing reports [8]. Among all detected constituents, thonningianin A had the highest content (86.32–99.38 mg/g DW), followed by kaempferol (9.87–10.68 mg/g DW), and pinocembrin-7-*O*-(3″-*O*-galloy-4″,6″-hexahydroxydiphenoyl)-glucoside (7.08–9.73 mg/g DW). The contents of gallic acid, (−)-epicatechin, rutin, isoquercitrin, kaempferol-3-*O*-rutinoside, and afzelin in all three samples were below 1 mg/g DW. Polyphenols are major functional constituents of *P. chinense* [22], and some reports also have shown evidence that polyphenols have a liver protective effect both in vitro and in vivo [23,24]. Thus, it is necessary to compare the biological activities of these samples in vitro.

### 3.4. Antioxidant Activity

Antioxidants are very important for human health, as they scavenge free radicals that can cause various diseases [25]. In this study, the antioxidant activities of UL, GTL, and BTL were determined by DDPH^·^, ABTS^·+^, and FRAP assays, and the IC_50_ values of DPPH^·^ and ABTS^·+^ were calculated using probit regression analysis. The inhibition rate of DPPH^·^ and ABTS^·+^ radicals were similar, with the scavenging activity of both decreasing in the following order: VE > GTL > BTL > UL (Figure 3). The maximum DPPH^·^ radical inhibition activity reached 87.73% (201 μg/mL), 87.30% (201 μg/mL), and 87.64% (201 μg/mL) for the extracts of GTL, BTL, and UL, respectively; while the maximum ABTS^·+^ radical inhibition activity reached 86.06% (67.12 μg/mL), 85.45% (100.46 μg/mL), and 87.64% (90.52 μg/mL) for the extracts of GTL, BTL, and UL, respectively. The extracts of GTL and BTL efficiently inhibited DPPH^·^ and ABTS^·+^ radicals at lower concentrations than those of UL. FRAP assays were performed to determine the transfer electron ability for the reduction of Fe^3+^ to Fe^2+^ in the presence of TPTZ [26]. Due to differences in reaction mechanisms, the absorbance values measured by FRAP assay exhibited dose-dependent growth trending in the order of VE > GTL > UL > BTL, indicating the reliability of the reducing power assay for different sample concentrations. 

The IC_50_ values of the DPPH^·^ and ABTS^·+^ radicals are shown in Figure 3. The differences in IC_50_ values among UL, BTL, and GTL reached significant levels (*p* < 0.05) in the following order: VE (17.41 μg/mL) < GTL (36.10 μg/mL) < BTL (39.99 μg/mL) < UL (45.46 μg/mL), demonstrating the potent in vitro antioxidant activity of *P. chinense* leaves after processing into green and black tea. For the IC_50_ values of ABTS^·+^, the order was similar to that of DPPH^·^, but the IC_50_ value of GTL (28.43 μg/mL) was slightly higher than that of VE (26.39 μg/mL, *p* > 0.05) and significantly lower than that of UL (37.99 μg/mL). In general, the antioxidant activity was directly related to the phenolic content [12]. Besides, the tea making involving enzymatic and non-enzymatic processes results in producing various active metabolites [27]. Therefore, the results of the DPPH^·^ and ABTS^·+^ indicated that the antioxidant activity of GTL and BTL were similar and higher than UL, and thus, GTL and BTL showed more promise as functional foods after processing.

### 3.5. ADH and ALDH Activities

Ethanol was metabolized through a two-step process in which ADH oxidizes ethanol to acetaldehyde, which was further oxidized to acetate by ALDH. Hence, ADH and ALDH are the principal enzymes responsible for the metabolism of ethanol in humans [14]. The effects of UL, GTL, and BTL on ADH and ALDH activities are shown in Figure 4. When the concentration of the samples was 25 μg/mL, all of them promoted ADH activity with the following order: GTL > UL > BTL (Figure 4A), while only UL and BTL could promote ALDH activity over a small range (Figure 4B). This boost in ALDH activity can mitigate the adverse effects of alcohol consumption by minimizing tissue exposure to acetaldehyde [28]. Both UL and GTL increased ADH and ALDH activities, while BTL increased ADH activity and decreased ALDH activity. The processing of green tea significantly improved ADH and ALDH activities compared with UL, while BTL promoted ADH activity, which is beneficial to the development of tea products derived from *P. chinense* leaves. Furthermore, the ADH activity showed a high positive correlation with the content of (−)-epicatechin (*R* = 0.945, *p* < 0.01) and afzelin (*R* = 0.938, *p* < 0.01) content as well as a high negative correlation with the content of kaempferol-3-*O*-rutinoside (*R* = 0.851, *p* < 0.05) and pinocembrin-7-*O*-(3″-*O*-galloy-4″,6″-hexahydroxydiphenoyl)-glucoside (*R* = 0.818, *p* < 0.05), whereas the ALDH activity showed a high positive correlation with the content of (−)-epicatechin (*R* = 0.945, *p* < 0.05), (+)-catechin (*R* = 0.912, *p* < 0.01), and afzelin (*R* = 0.948, *p* < 0.01). The effect of polyphenol on ADH and ALDH activities includes two mechanisms: non-covalent binding [29] and ionic action [30]. Studies have reported that (−)-epicatechin, (+)-catechin, and pinocembrin-7-*O*-[3′′-*O*-galloyl-4′′,6′′-hexahydroxydiphenoyl]-β-D-glucose may be the active ingredients in *P. chinense* for the observed hepatoprotective effects [7,31].

## 4. Conclusions

This work comprehensively revealed the differences in volatile compounds, phenolic constituents, and biological activities of *P. chinense* leaves and their tea products. Through the analysis of *P. chinense* leaves, 43 volatile compounds in UL, GTL, and BTL were identified as differential metabolites. The TPC, TFC, and TPAC were all significantly higher in UL than in GTL and BTL. Notably, 14 phenolic compounds were determined, of which pinocembrin 7-*O*-(3″-*O*-galloy-4″,6″hexahydroxydiphenoyl)-glucoside and pinocembrin content were significantly higher in BTL and GTL than in UL. GTL also showed higher antioxidant, ADH, and ALDH activities than UL. Above all, processing the leaves into green tea not only increased the active metabolites contents than raw leaves but also significantly improve biological activity (antioxidant and hepatoprotective activity). Therefore, *P. chinense* leaves were promising materials with high economic benefits and its green tea product could be developed into a healthy product for daily consumption for soothing and nourishing the liver.

## Figures and Tables

**Figure 1 foods-13-00399-f001:**
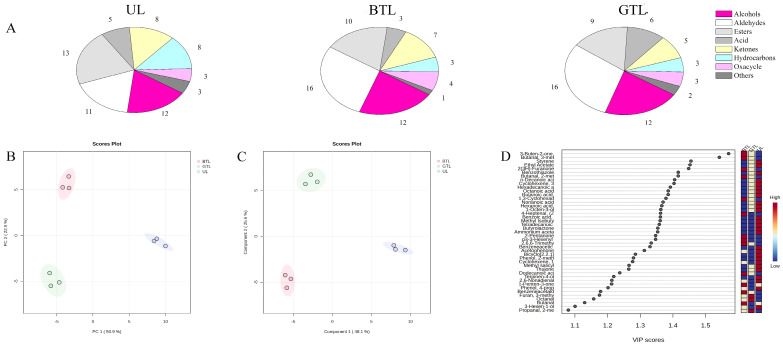
Volatile components of *P. chinense* leaves (UL), *P. chinense* leaves green tea (GTL) and *P. chinense* leaves black tea (BTL). (**A**) The number and composition of the volatile flavor substances in UL, GTL, and BTL. (**B**) Principal component analysis (PCA) score plot for volatile metabolites. (**C**) Partial least squares discriminant analysis (PLS-DA) score plot for volatile metabolites. (**D**) Variables important in projection (VIP) of volatile components.

**Figure 2 foods-13-00399-f002:**
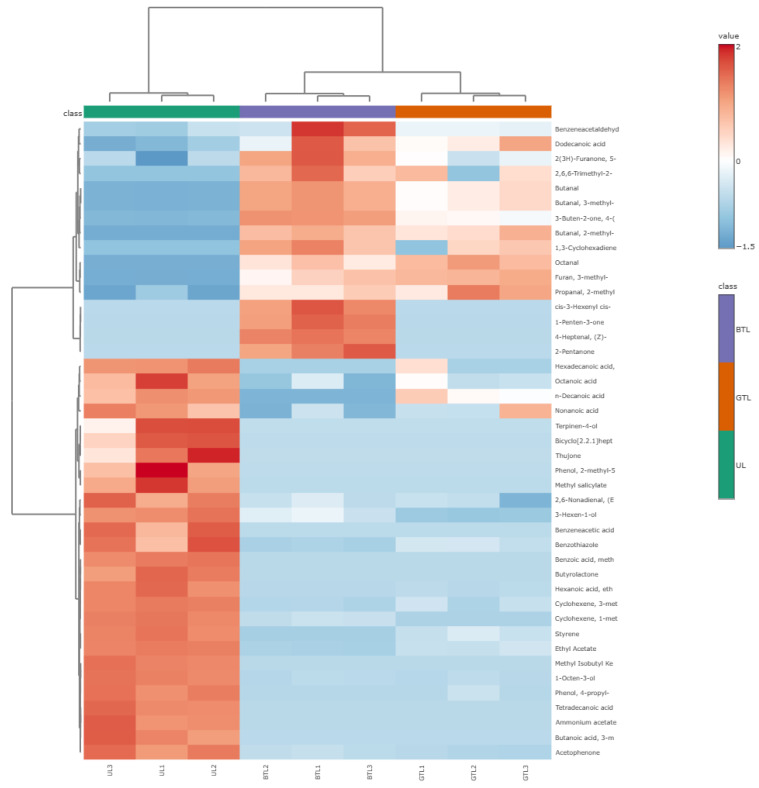
Heat map of the contents of significantly differential volatile components in *P. chinense* leaves (UL), *P. chinense* leaves green tea (GTL), and *P. chinense* leaves black tea (BTL).

**Figure 3 foods-13-00399-f003:**
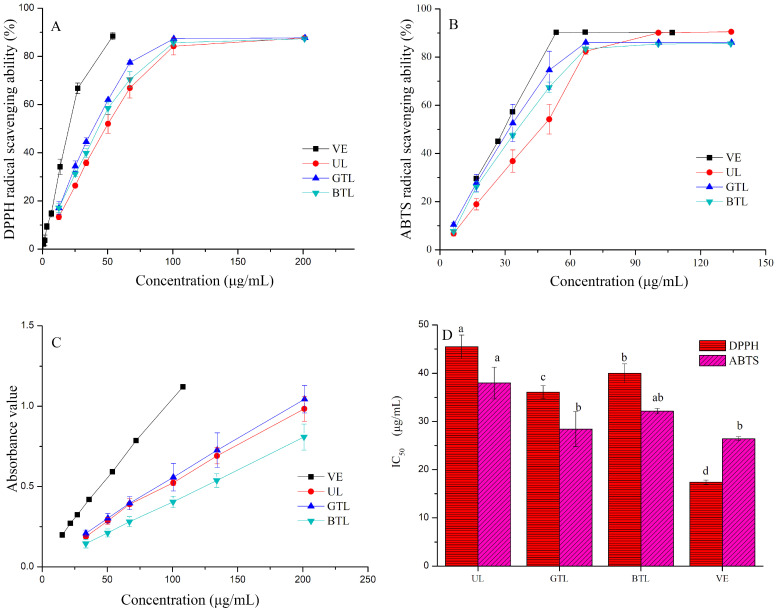
The DPPH^·^ (**A**), ABTS^·+^ (**B**) radical scavenging values, ferric reducing power (FRAP, (**C**)) and (**D**) the IC_50_ values of VE, *P. chinense* leaves (UL), *P. chinense* leaves green tea (GTL), and *P. chinense* leaves black tea (BTL).

**Figure 4 foods-13-00399-f004:**
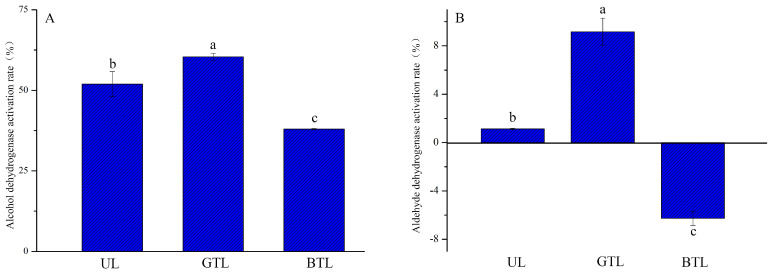
The alcohol dehydrogenase (ADH) activity (**A**) and acetaldehyde dehydrogenase (ALDH) activity (**B**) of *P. chinense* leaves (UL), *P. chinense* leaves green tea (GTL) and *P. chinense* leaves black tea (BTL).

**Table 1 foods-13-00399-t001:** Total phenolic content (TPC), total flavonoid content (TFC), and total proanthocyanidins content (TPAC) of three *P. chinense* raw leaves (UL), green (GTL), and black tea (BTL).

	UL	GTL	BTL
TPC (mg GAE/g DW)	179.34 ± 12.53 ^a^	138.90 ± 6.98 ^b^	97.13 ± 7.32 ^c^
TFC (mg RE/g DW)	71.93 ± 0.52 ^a^	59.88 ± 0.83 ^b^	40.07 ± 2.53 ^c^
TPAC (mg CE/g DW)	65.91 ± 0.99 ^a^	57.74 ± 0.42 ^b^	54.13 ± 2.14 ^c^

^a–c^ Data with different superscript lowercase letters in the same row were significantly different (*p* < 0.05).

**Table 2 foods-13-00399-t002:** The contents of phenolic compounds of three *P. chinense* raw leaves (UL), green (GTL), and black tea (BTL) (mg/g dw).

Compounds	UL	GTL	BTL
gallic acid	0.44 ± 0.01 ^a^	0.22 ± 0.01 ^c^	0.32 ± 0.01 ^b^
(+)-catechin	5.91 ± 0.39 ^ab^	6.68 ± 0.48 ^a^	5.21 ± 0.11 ^b^
(−)-epicatechin	0.26 ± 0.03 ^a^	0.27 ± 0.03 ^a^	-
rutin	0.06 ± 0.01 ^a^	0.07 ± 0.01 ^a^	0.06 ± 0.00 ^a^
isoquercitrin	0.22 ± 0.02 ^a^	0.23 ± 0.02 ^a^	0.21 ± 0.01 ^a^
kaempferol-3-*O*-rutinoside	0.18 ± 0.01 ^a^	0.18 ± 0.01 ^a^	0.20 ± 0.00 ^a^
astragaline	1.61 ± 0.18 ^a^	1.78 ± 0.17 ^a^	1.83 ± 0.05 ^a^
quercetin	4.88 ± 0.53 ^a^	5.36 ± 0.52 ^a^	5.48 ± 0.14 ^a^
afzelin	0.88 ± 0.01 ^a^	0.91 ± 0.00 ^a^	0.82 ± 0.01 ^b^
pinocembrin 7-O-glucoside	2.70 ± 0.02 ^a^	0.23 ± 0.02 ^b^	0.28 ± 0.01 ^b^
kaempferol	10.46 ± 0.55 ^a^	10.68 ± 0.00 ^a^	9.87 ± 0.60 ^a^
pinocembrin 7-*O*-(3″-*O*-galloy-4″,6″-hexahydroxydiphenoyl)-glucoside	7.08 ± 0.81 ^b^	7.69 ± 0.86 ^b^	9.73 ± 0.41 ^a^
pinocembrin	0.76 ± 0.12 ^b^	1.20 ± 0.09 ^a^	0.93 ± 0.05 ^b^
thonningianin A	86.32 ± 6.33 ^a^	99.38 ± 9.04 ^a^	96.30 ± 0.26 ^a^

^a–c^ Data with different superscript lowercase letters in the same row were significantly different (*p* < 0.05); - means not be detected.

## Data Availability

Data are contained within the article and Appendix A.

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
