# Peer review of "Comprehensive Analysis of Phenolic Constituents, Biological Activities, and Derived Aroma Differences of Penthorum chinense Pursh Leaves after Processing into Green and Black Tea"

_foods, 2024, doi:10.3390/foods13030399_

Round 1
Reviewer 1 Report
Comments and Suggestions for Authors
This study determines the effect of Penthorum chinese leaves processing into green and black tea on the volatile compounds profile, polyphenols contents, as well as in vitro antioxidant and hepatoprotective potential. The use of edible ethnobotanical sources of bioactive compounds and their development by processing is in accordance with current trends in food science and diet prevention issues, therefore the topic of this study is quite interesting.
The abstract is well-written. The introduction provides a good background of the study and includes relevant references. The experiments are well designed and described; however, I recommended changing the sequence of subsections to make it more ordered and following the presented results i.e. 2.1. Chemicals and reagents; 2.2. Preparation of tea samples; 2.3. Analysis of volatile components (because these results are presented in the first subsection of Results and Discussion); 2.3. Sample extraction… etc. or change the order of subsections in Results and Discussion. The results description is clear, but Figures 1 and 2 should be much more enlarged. In the current version, they are difficult to read. The discussion is supported by the results. The conclusion summarizes the most important findings, however, it contains an artefact of the manuscript template, which must be removed „This section is not mandatory but can be added to the manuscript if the discussion is unusually long or complex.”
Due to some described shortcomings of the “technical” nature, I recommended minor revision.
Reviewer 2 Report
Comments and Suggestions for Authors
General comment
The manuscript ''Comprehensive analysis of phenolic constituents, biological activities, and derived aroma differences of Penthorum chinense Pursh leaves after processing into green and black tea'' described content of volatile and phenolic compounds in untreated leaves as well as in green tea leaves and black tea leaves of Penthorum chinense (family Penthoraceae). Additionally, the antioxidant, alcohol dehydrogenase and acetaldehyde dehydrogenase activities were studied. The results showed that 62, 57, and 56 volatile compounds were detected in untreated leaves, green tea leaves and black tea leaves, respectively. The similar trends were noticed in total phenolic (97.13–179.34 mg GAE/g DW), flavonoid (40.07–71.93 mg RE/g DW), and proanthocyanidin (54.13–65.91mg CE/g DW) content. Fourteen phenolic compounds were determined
Comments
Abstract
Line 14: add family ''Penthoraceae'' after ''Purs''
1. Introduction
Line 36: put ''Pursh'' in Normal and add ''family Penthoraceae'' after ''Purs''
Line 53: explain abbreviation ''anti-HBV'' (it is mentioned for the first time)
Lines 55, 56: add Latin names (in brackets) after common name of mentioned species ''mulberry.., Dang Shen…, and mango''
2. Materials and Methods
Line 64: add details of collection site (GPS coordinates, altitude) and deposited voucher material (Herbarium abbreviation, depostion place...). These data are ordinary part of scientific article.
Line 66: with or without space before symbol for Celsius degree, not both (compare with style in lines 67-77, 91, 110, 124, 126, 131, 150…). Please, check all manuscript.
Paragraph with description of used statistical methods should be added.
3. Results and Discussion
Line 174: add Latin names of plant after ''Kombucha tea''
Figure 1 is unreadable. Please, separate images A from other, as well B and C from D. Also, show identified volatile compounds in UL, GTL and BTL in form of table (it is more adequate way of presented results). Figure 1A and results of PCA score analysis and PLS-DA should stay but only in a much higher resolution.
Line 221: put ''P. chinense'' in Italic
Line 241: put ''proanthocyanidins'' in Normal
Line 301: put ''P. chinense'' in Italic
Line 328: put ''P. chinense'' in Italic
4. Conclusions
Lines 331-332: omit ''This section is not mandatory but can be added to the manuscript if the discussion is 331 unusually long or complex.''
Lines 336-337: change font size in ''compounds were determined''
References
The references are not arranged according to the journal style. Also, the journal names should be written on the same way (for examole, compare ref. no 2 and 3). So, see the journal style and make some correctins.
Line 374: change ''penthorum'' into ''Penthorum''
Line 378: put ''Eurotium cristatum'' into Italic
Line 83: omit do and space before ''Mango''
Lines 400-401: use abbreviation instead of ''Chemometrics and 400 Intelligent Laboratory Systems''
Line 426: put ''Penthorum chinense'' into Italic
Reviewer 3 Report
Comments and Suggestions for Authors
The manuscript studies the effect of different processing methods of tea products made from Penthorum chinense Pursh leaves on their phenolic content, aroma compounds and biological activity, The aim of the study is not clearly and sufficiently highlighted as well as the applications of the research and how the conclusions of the study could provide a different approach on tea preparation and consumption trends.
Therefore, my recommendation is that the aim of the study as well as the conclusions section should be revised accordingly with the final scope to provide new insights and novelty on the research topic.
Below authors will find detailed comments:
Introduction: Detailed information should be provided on different preparation methods of tea products.
2.1 Preparation of tea samples:
What is the purpose of treating differently green tea leaves and black tea leaves? reasons of eliminating toxicity??
2.10 ADH and ALDH activities
L.159-161: Authors should explain better how does Q (activation rate) come about.
Results and Discussion:
L.184: BTL had a higher aroma concentration than ...than what??
Figures 1 and 2 should be provided in better resolution. They are illegible.
L. 258-260: Authors should discuss briefly their findings and how they are related or not with other research studies.
3.4 Antioxidant activity:
Throughout the paragraph a new acronym (VE) was inserted. VE stands for what??
L.296: Authors could compute the Pearson's correlation coefficient.
Conclusions
L.331-332: Delete the sentence.
Comments on the Quality of English Language
Minor editing of English language is required.
Reviewer 4 Report
Comments and Suggestions for Authors
I send a review of manuscript ID number foods-2798953, of the authors: Zhuoya Xiang, Boyu Zhu, Xing Yang, Junlin Deng, Yongqing Zhu, Lu Gan, Manyou Yu, Jian Chen, Chen Xia, Song Chen „Comprehensive analysis of phenolic constituents, biological activities, and derived aroma differences of Penthorum chinense Pursh leaves after processing into green and black tea”.
I think that the manuscript deals with an interesting area of scientific research the question of analysis of phenolic constituents contents, biological activities, and volatile components of Penthorum chinense Pursh leaves after their processing into green and black tea. I think that the authors should make a minor revision.
Generally, the manuscript contained words written in too small a font, e.g. page 6, line 241, page 9, lines 336-337, etc., and text in a different color, e.g. page 7, lines 294-288, page 8, lines 315- 323, etc.
- Introduction:
Page 2, lines 45-61; I request the Authors to provide more convincing arguments of the advisability of undertaking the presented research. I would like to ask the Authors to formulate the purpose of the paper precisely and clearly. Since in this form it is not clear and convincing to the Reviewer.
- Materials and methods:
Page 2, lines 73; …pan-fired at 320 °C? Whether pan-frying was used in the study at this temperature, is not clear?
Page 4, line 161 - 2.11. Statistical analysis? I would like to ask the Authors to provide the missing information on the statistical analysis of the results obtained in the study. Please also provide information on how many replicates of each analysis were performed? Please provide information on the standard deviation (±SD) and number of repetitions (n=) in the appropriate place under the Tables and Figures.
4. Conclusions
Page 9, lines 331-332 - This section is not mandatory but can be added to the manuscript if the discussion is 331 unusually long or complex. It is not necessary to include this sentence in the conclusions - please remove it. Could the Authors indicate one most recommended (most beneficial) product in their conclusions?
References
All literature is cited in the text.
Ps. I wish you a Happy New Year.
Round 2
Reviewer 2 Report
Comments and Suggestions for Authors
Dear Authors,
I have no other comments. I hope that the resolution of Figure 1 will be enough.
Author Response
Thank you for your comments. We have provided the higher resolution of Figure 1 in Figure RAR.
Reviewer 3 Report
Comments and Suggestions for Authors
The manuscript was revised in some points; however, still the aim of the study as well as the conclusions section was not revised, as suggested, with the final scope to provide new insights and novelty on the research topic.
Author Response
Thank you for your suggestion and we revised the aim of the study in Iintroduction section and conclusions section in red part.